# Pre-Compensation of Mold in Precision Glass Molding Based on Mathematical Analysis

**DOI:** 10.3390/mi11121069

**Published:** 2020-11-30

**Authors:** Yue Zhang, Kaiyuan You, Fengzhou Fang

**Affiliations:** 1State Key Laboratory of Precision Measuring Technology & Instruments, Centre of Micro/Nano Manufacturing Technology–MNMT, Tianjin University, Tianjin 300072, China; yuede1993@163.com (Y.Z.); youkaiyuan@tju.edu.cn (K.Y.); 2Centre of Micro/Nano Manufacturing Technology–MNMT-Dublin, University College Dublin, 4 D04 V1W8 Dublin, Ireland

**Keywords:** precision glass molding, mold pre-compensation, mathematical analysis, cooling rate, surface form error

## Abstract

Precision glass molding is the most appropriate method for batch production of glass lenses with high surface accuracy and qualified optical performance. However, the form error caused by material expansion and contraction is the main factor affecting the precision of the molded lenses, thus the mold must be pre-compensated. In this paper, an effective method of mold pre-compensation based on mathematical analysis is established. Based on the thermal expansion curve of D-ZK3 glass, the freezing fictive temperature of the glass under the actual cooling rate is measured, and the mold pre-compensation factor can be quickly calculated. Experimental results show that the peak valley (PV) value of the surface form error of molded aspheric lens with an aperture of 5.3 mm is effectively reduced from 2.04 μm to 0.31 μm after the pre-compensation, thus meeting the geometric evaluation criterion.

## 1. Introduction

With the development of science and technology, glass lenses find wide applications in various fields such as aerospace, laser projection, biomedical and consumer electronics. The demands for glass lenses with higher surface accuracy and better optical performance are also increasing. While the production efficiency of traditional glass grinding and polishing cannot meet the increasing demand of glass lenses, the precision glass molding (PGM) technology has become an efficient processing technology to meet the urgent requirement. PGM has become a research hotspot in the industrial and academic fields because it has the advantages of environmental friendliness, high efficiency, low cost and mass production [1,2,3,4,5,6].

PGM is a kind of processing technology, which uses precision molds with a specific design surface to press the glass preform at an elevated temperature so that the mold surface form can be duplicated on the lenses [7]. However, the form error of molded glass lens caused by material expansion and contraction seriously affects its accuracy and performance, thus the mold must be compensated. The traditional trial-and-error method of mold compensation often requires the mold to be processed many times, reducing the efficiency and causing extra production cost. Therefore, it is necessary to pre-compensate the mold, and the molded lenses can meet the geometric evaluation criterion through only one-time processing of the mold.

Mold pre-compensation aims to estimate the amount of compensation according to the ideal surface profile of the lens, and then obtain the compensated mold surface profile. Finite elements analysis (FEA) is the most commonly used method for mold pre-compensation in current research [8,9,10]. Yi et al. [11] simulated the PGM process of aspheric lens through FEA, and described the surface profile deviation of the molded lens with the mold, which was consistent with the experimental results on the trend. Jain [12] studied the distribution of the residual stress involved in molded glass lens through FEA simulation, and analyzed the influence of glass thermal expansion coefficient, molding speed, molding temperature and cooling rate on lens surface profile deviation. Wang et al. [13] predicted the curve change of molded aspheric glass lens by FEA. Additionally, the difference between the simulation result and the experimental result was about 2 μm. The surface form error was reduced from 12 μm to less than 1 μm by the mold pre-compensation method. Zhou et al. [14,15] used commercial FEA software to conduct the numerical simulation of the whole PGM process, and the influence of the holding force on the lens profile deviation during the annealing stage was studied. The maximum surface profile deviation of the molded lens was reduced to about 0.04 μm through several mold pre-compensation iterations by FEA, but there was no experimental verification. Su et al. [16] proposed a comprehensive mold pre-compensation method, which can simultaneously compensate for refractive index change and profile deviation. An aspheric glass lens was molded and the geometry variation was compensated to be less than 1.5 μm.

FEA can be employed to qualitatively analyze the trend of surface form deviation, but its accuracy is completely dependent on the accuracy of the FEA mold, which requires very complicated glass constitutive model measurement and system parameters checking. There is often an obvious error between the simulation results and the experimental results [13]. Therefore, there is an urgent need to develop a convenient and effective method of mold pre-compensation for PGM. In this study, a method of mold pre-compensation based on mathematical analysis is established.

## 2. Principle and Method of Mold Pre-Compensation

### 2.1. Mold Compensation Principle

#### 2.1.1. Material Expansion and Contraction

Expansion and contraction mean that most materials expand when the temperature increases and contract when the temperature decreases. The essence of expansion and contraction of materials can be attributed to the fact that the average distance between particles in lattice structure is positively correlated with temperature, which comes from the anharmonic motion of atoms. As shown in Figure 1a, the resultant force on the particle at the equilibrium position (the distance between the adjacent particles is *r*_0_) is 0. With an increase in temperature, the vibration amplitude of the particle increases. Because the resultant force on both sides of the particle at the equilibrium position is asymmetric, when *r* < *r*_0_, the repulsion force increases rapidly; when *r* > *r*_0_, the attraction force increases slowly. Therefore, its equilibrium position moves to the right, and the average distance between adjacent particles increases.

Figure 1b is a schematic diagram of isotropic materials expansion in heat. In the initial case, the average distance between adjacent particles is *r*_0_, and when the temperature changes Δ*T*, the average distance goes to *r*:(1)$$r=(1+\alpha \Delta T)r_{0},$$
where *α* is the average linear expansion coefficient of the material. Similarly, the equation is also applicable to material contraction.

For PGM, both glass and mold are isotropic materials and the linear expansion coefficient in all directions is consistent. Figure 2a,b are respectively schematic diagrams of mold and glass expanding and contracting in the PGM process. As the mold and the glass lens are rotationally symmetric, only the half section is analyzed. In order to show the variation trend of expansion and contraction of materials more obviously, the linear expansion coefficients larger than the actual materials themselves are adopted in FEA simulation. Considering that the glass lens is always in contact with the lower mold surface due to gravity, the aspheric surface vertex is set as the fixed contact point limiting the degree of freedom of displacement. 

The PGM process can be divided into four stages: heating and soaking, molding, annealing and rapid cooling. Firstly, in the heating and soaking stage, the mold and glass preform are heated from room temperature to molding temperature, and their volume becomes larger. In the molding stage, the temperature of the mold and the glass remains stable, the glass preform deforms under pressure, and the profile of the lens is consistent with the high temperature mold. Furthermore, in the annealing stage, the temperature decreases slowly and the mold and lens volume decreases. It should be noted that the glass is viscoelastic in the transition region, and the structure relaxation leads to the nonlinear change of its volume which is related to the rate of cooling. Finally, in the rapid cooling stage, the mold and glass can be approximated as elastic materials, and the linear expansion coefficient can be regarded as constant.

#### 2.1.2. Glass and Mold Contraction during the Cooling Stage in PGM

In the process of PGM, the mold can be regarded as elastic material, its relative linear expansion and contraction are proportional to the temperature change, and the proportional coefficient is the average linear expansion coefficient *α_M_*.

As for the glass, due to its viscoelastic characteristic in the transition region, the structure relaxation leads to the change of glass structure lagging behind the change of temperature, which results in the nonlinear change of glass volume. The degree to which the glass structure lags behind equilibrium state is usually characterized by fictive temperature *T_f_*. The fictive temperature is a pure mathematical quantity that quantifies the actual structural state of glass at a certain temperature. The cooling rate in the PGM process is the most important factor to determine the structure relaxation performance of the glass. Figure 3 shows the fictive temperature and the volume change of the glass under different cooling rates during the cooling stage of PGM [14,15].

As shown in Figure 3a, the fictive temperature *T_f_* of the equilibrium liquid glass is the same as the actual temperature *T*; in the glass transition region, *T_f_* of the viscoelastic state glass is higher than *T*; and *T_f_* of the glass in the glassy state tends to be stable and finally stabilizes at the freezing fictive temperature *T_F_*. Besides, the slower the cooling rate is, the lower the corresponding *T_f_* and *T_F_* are.

As shown in Figure 3b, the volume change caused by cooling contraction is also affected by the cooling rate. The slower the cooling rate is, the smaller the glass volume is, and the larger the volume change is. For the convenience of calculation, the glass contraction during the cooling stage can be divided into two approximate steps: contracting with liquid volume thermal expansion coefficient *α_vl_* from the molding temperature *T_M_* to the freezing fictive temperature *T_F_*; and contracting with solid volume thermal expansion coefficient *α_vg_* from freezing fictive temperature *T_F_* to the room temperature *T_R_*. The same is true for the linear thermal expansion coefficients *α_l_* and *α_g_*.

#### 2.1.3. Mold Pre-Compensation

In the process of PGM, due to the difference between the thermal expansion coefficient of the mold and the glass, there would be a gap between the molded lens and mold, leading to the difference between the surface profiles, resulting in surface form error of molded lens.

Figure 4 shows the schematic diagram of the mold and lens surface profiles change in the PGM process. The abscissa is the radial distance *x*, the ordinate is the aspheric surface profile *z*, and the origin is the contact point between the lower mold and the glass. The gray curve is the mold profile at room temperature. The red curve is the mold profile at molding temperature, that is, the lens profile at molding temperature. The blue curve is the final profile of the molded glass lens after cooling to room temperature.

As shown in Figure 4a, without any compensation, the mold at room temperature is processed into the ideal lens profile as shown by the gray curve. After the temperature rises to the molding temperature and the molding stage is completed, due to the thermal expansion, the profiles of the mold and lens change to the red curve. In the cooling stage, as the temperature decreases to room temperature, the mold profile returns to the gray curve, and the molded lens profile changes to the blue curve. At this time, there is a gap between the final profile of the molded lens and the ideal profile, and the shaded part in the figure is the surface form error of the molded lens. In order to reduce the surface form error of the molded lens and make its final profile closer to the ideal one, it is necessary to compensate for the profile of the mold at room temperature.

As shown in Figure 4b, the compensated mold profile at room temperature is shown by the gray curve, and the shaded part is the compensation value based on the ideal lens profile. The profiles of the mold and lens at molding temperature are shown as the red curve. After cooling, the mold profile returns to the gray curve, while the final profile of the molded lens changes to the blue curve, that is, the ideal lens profile. By means of mold compensation, the surface form error of the molded lens can be greatly reduced, and finally meet the geometrical criterion.

### 2.2. Mold Pre-Compensation Based on Mathematical Analysis

The key to mold pre-compensation is to estimate the amount of expansion and contraction of the mold and lens during the PGM process. In order to make the mold pre-compensation process more convenient, a mold pre-compensation method based on mathematical analysis is established. In this method, the glass and mold are regarded as isotropic materials, and the ratio of expansion and contraction of the material are calculated in combination with the PGM technical parameters. The aspheric surface expression is directly compensated according to the logic diagram as shown in Figure 5.

In the forward PGM process, the mold is heated to molding temperature from room temperature, and the relative linear expansion is *TE_M_*. After the molding stage, the surface profile of the lens is consistent with that of the mold at the molding temperature. Subsequently, the lens is cooled and contracted, and the relative linear contraction of lens is *TE_G_*.

In the backward mold pre-compensation process, the final surface profile of the molded lens at room temperature is set as the known ideal aspheric surface expression *z*_1_(*x*). Then, the intermediate aspheric surface expression *z*_2_(*x*) can be obtained by bidirectional expansion in proportion with the relative linear expansion of *TE_G_*. Finally, the compensated aspheric surface expression of the mold at room temperature *z*_3_(*x*) can be obtained from the intermediate aspheric surface by bidirectional contraction in proportion with the relative linear contraction of *TE_M_*.

The expression of ideal aspheric lens *z*_1_(*x*) is written as the standard form of universal even-order aspheric surface:(2)$$z_{1}(x)=\frac{x^{2}}{R_{1}(1+\sqrt{1-(1+k)\frac{x^{2}}{{R_{1}}^{2}}})}+A_{2n}x^{2n}(n=1,2,3,4\ldots )$$
where *R*_1_ is the radius of curvature at the apex of the aspheric surface, *k* is the conic constant, and *A*_2*n*_ is the high-order coefficient of the aspheric surface. In the *x* and *z* directions, expansion in proportion is carried out by the scaling factor (1+*TE_G_*), and the intermediate aspheric expression *z*_2_(*x*) is obtained:(3)$$z_{2}(x)=(1+TE_{G})\frac{{\left(\frac{x}{1+TE_{G}}\right)}^{2}}{R_{1}(1+\sqrt{1-(1+k)\frac{x^{2}}{{R_{1}}^{2}}}{\left(\frac{x}{1+TE_{G}}\right)}^{2})}+(1+TE_{G})A_{2n}{\left(\frac{x}{1+TE_{G}}\right)}^{2n}$$

Then, based on the intermediate aspheric expression *z*_2_(*x*), contraction in the *x* and *z* directions is carried out by *TE_M_*. In other words, contraction in proportion is carried out by the scaling factor (1−*TE_M_*), and the compensated aspheric expression *z*_3_(*x*) is obtained:(4)$$z_{3}(x)=\frac{\frac{x^{2}}{R_{1}(1+TE_{G})(1-TE_{M})}}{\left(1+\sqrt{1-(1+k)\frac{x^{2}}{{R_{1}}^{2}{\left(1+TE_{G}\right)}^{2}{\left(1-TE_{M}\right)}^{2}}}\right)}+\frac{A_{2n}}{{\left(1+TE_{G}\right)}^{2n-1}{\left(1-TE_{M}\right)}^{2n-1}}x^{2n}$$

By comparing the compensated aspheric expression *z*_3_(*x*) with the standard form of universal even-order aspheric surface, it can be obtained as follows:(5)$$z_{3}(x)=\frac{x^{2}}{R_{3}(1+\sqrt{1-(1+k)\frac{x^{2}}{{R_{3}}^{2}}})}+{A^{\prime }}_{2n}x^{2n}$$
(6)$$R_{3}=(1+\gamma )R_{1}$$
(7)$${A^{\prime }}_{2n}=\frac{1}{{\left(1+\gamma \right)}^{2n-1}}A_{2n^{\prime }}$$
(8)$$\gamma =(1+TE_{G})(1-TE_{M})-1$$
where *γ* is defined as the pre-compensated scaling factor.

In the PGM process, it is necessary to estimate the relative linear thermal deformation of the mold *TE_M_* and lens *TE_G_* in combination with the actual technical parameters. The changes of the aspheric surface profile of the mold and glass lens are considered respectively under the condition of cooling without holding pressure and cooling with holding pressure. In the case of cooling without holding pressure, the glass lens contracted freely from *T_M_* to *T_R_*. The effective relative linear thermal deformations of the mold and lens are:(9)$$TE_{M}=\alpha_{M}(T_{M}-T_{R})$$
and
(10)$$TE_{G}=\alpha_{1}(T_{M}-T_{F})+\alpha_{g}(T_{F}-T_{R})$$

Respectively.

Under the condition of cooling with holding pressure, a holding pressure is still applied to the mold during the slow cooling stage after the molding stage so as to ensure that the glass lens is always attached to the aspheric surface of the mold. After the end of the slow cooling stage, the holding pressure is released and the temperature drops rapidly to *T_R_*, and the glass lens contracts freely. Take the aspheric surface at *T_F_* as the intermediate aspheric surface, and the effective relative linear thermal deformations of the mold and lens are:(11)$$TE_{M}=\alpha_{M}(T_{F}-T_{R})$$
and
(12)$$TE_{G}=\alpha_{g}(T_{F}-T_{R})$$
respectively.

According to Equations (5) to (12), the compensated aspheric expression of the mold can be directly obtained based on the ideal aspheric surface expression and actual technical parameters in the PGM process.

## 3. Experiments and Result Discussion

### 3.1. Thermal Expansion Curve

The nominal glass transition temperature *T_g_* can be regarded as the freezing fictive temperature *T_F_* when the glass preform is cooled at a very slow rate during fabrication. However, considering the production efficiency, the cooling rate in the actual PGM process is much higher than that in the glass preform manufacturing process, so it is necessary to measure the actual *T_F_*.

As shown in Figure 6a, the thermal dilatometer (NETZSCH’s DIL 402 Expedis Classic) was used for the thermal expansion curve test. Figure 6b shows the principle of measurement. When the glass sample expands and contracts during the test, the green parts in the figure move backward under the guidance of the blue linear guide, and the corresponding length changes can be measured and recorded by the optical encoder.

Dense barium crown optical glass D-ZK3 was selected as the typical low melting point glass, and the thermal expansion curve within the range from *T_R_* to *T_M_* is measured. The glass sample was processed into a cylinder with a diameter of 8 mm and a length of 25 mm, and the two end surfaces were polished. According to the actual conditions of PGM, the molding temperature and cooling rate were set to 550 °C and 0.2 K/s, respectively. The measurement of thermal expansion curve can be divided into two stages. Firstly, the temperature rises from *T_R_* to 550 °C with a heating rate of 4 K/min based on test criteria. After that, the temperature decreases from 550 °C to *T_R_*, and the cooling rate is controlled at about 0.2 K/s. Additionally, the freezing fictive temperature of the glass under the specific cooling rate can be obtained by lengthening the straight lines of the low temperature region and high temperature region on the measured thermal expansion curve.

Figure 7 shows the measurement result of the thermal expansion curve. The curve from green dot to yellow dot represents the heating stage, while the curve from yellow dot to blue dot represents the cooling stage. In the heating stage curve, the glass transition temperature *T_g_* is 510 °C obtained by the drawing method, which is consistent with the nominal value (511 °C). In the cooling stage curve, the freezing fictive temperature *T_F_* is 522 °C obtained by the drawing method. In addition, it can be obtained that the average solid linear thermal expansion coefficient of glass *α_g_* is about 10.3 × 10^−6^ /K and the average liquid linear thermal expansion coefficient *α_l_* is about 3.3 × 10^−5^ /K.

### 3.2. Mold Pre-Compensation

Take the plane-convex aspheric glass lens as an example, the aspheric surface of the mold is pre-compensated. The aspheric coefficients of the ideal lens are listed in Table 1, and the effective diameter is 5.3 mm.

The mold material is tungsten carbide (RCCFN) and the thermal expansion coefficient is 4.9 × 10^−6^ /K. When considering the PGM process with holding pressure which is the most common mode, according to Equations (5)–(12), the calculated value of the pre-compensated scaling factor *γ* is 0.002698, and the aspheric coefficients of the compensated mold are listed in Table 2.

In order to compare and verify the effect of mold pre-compensation, two molds were machined by ultra-precision grinding according to the aspheric coefficients of the ideal lens and compensated mold, respectively. The molds were machined based on the ultra-precision lathe (650FG, Moore Nanotechnology System). The normal single point grinding method based on B axis was used [17,18,19,20]. The resin binder cylindrical diamond grinding wheel with rounded corner of #325 and #2000 was respectively used for crude grinding and fine grinding of the molds. The specific machining parameters are listed in Table 3.

The two molds were used to conduct the PGM experiments on a single station glass molding machine (GMP-311V, Toshiba, Tokyo, Japan) as shown in Figure 8a. The PGM process conditions are shown in Figure 8b. It should be noted that in the PGM process, the cooling rate was set to 0.2 K/s, and the holding force was set to 0.2 kN during the annealing stage. The two groups of experiments had the same process conditions except that the aspheric coefficients of the molds are different. The fine grinding mold used in PGM experiment and molded glass lens are shown in Figure 9a,b, respectively.

### 3.3. Form Error of the Molded Lenses

A surface profilometer (Talysurf PGI Optics 840, Taylor Hobson, Leicester, UK) was used to measure the aspheric surface profile of the glass lenses molded by two sets of molds. Since the aspheric surface is centrally symmetric, only one meridian passing through its vertex needs to be measured. Based on the aspheric coefficients of the ideal lens, the surface form errors of the actual molded glass lenses were evaluated. The measurement results of form error of glass lenses molded by uncompensated and pre-compensated molds are shown in Figure 10a,b, respectively.

The measurement results show that the degree of the aspheric surface profile of molded lens deviating from the ideal surface profile increases with the increase of the distance from the center. For the lens with an effective aperture of 5.3 mm in this case, the peak valley (PV) value of the aspheric surface form error of the glass lens molded by the uncompensated mold is 2.04 μm, which cannot meet the geometric evaluation criterion. After the pre-compensation, the PV value of the aspheric surface form error is reduced to 0.31 μm, which is acceptable. Therefore, the mold pre-compensation method of PGM based on mathematical analysis can effectively optimize the aspheric surface form error and improve the quality of the molded glass lens.

## 4. Conclusions

In this paper, the thermal deformation trend of the mold and glass at different stages in the PGM process are presented. The problem that the aspheric surface form error of molded glass lens cannot meet the geometric evaluation criterion, caused by the difference of thermal expansion coefficient between glass and mold material, is analyzed. 

Compared with the traditional mold pre-compensation method based on FEA, the mold pre-compensation method based on mathematical analysis can obtain the compensated aspheric expression of the mold directly by the aspheric expression of the ideal lens without complicated glass constitutive model measurement and system parameters checking which is hard to do in actual production. For the typical plane-convex aspheric glass lens with an effective aperture of 5.3 mm, the PV value of the form error of molded lens is reduced from 2.04 μm to 0.31 μm by mold pre-compensation, so as to meet the geometric evaluation criterion. The experiment results show that the mold pre-compensation method based on mathematical analysis can optimize the surface form error of molded glass lenses quickly and effectively improve machining quality.

The influence of more kinds of PGM-process parameters will be studied to further improve the accuracy of mold pre-compensation in future work. Meanwhile, the effect of more kinds of surface profiles will also be investigated to improve the comprehensiveness of the mold pre-compensation method.

## Figures and Tables

**Figure 1 micromachines-11-01069-f001:**
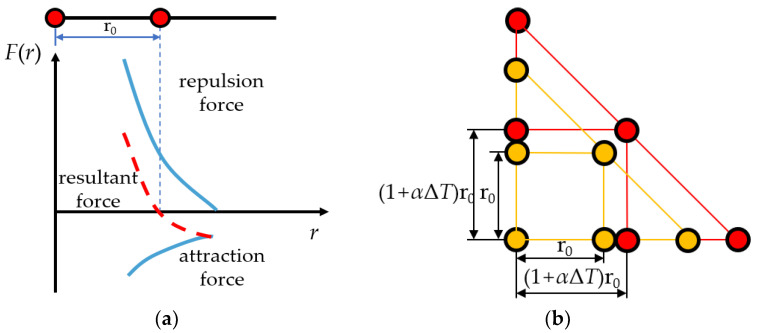
Schematic diagram of expansion and contraction. (**a**) The principle of expansion and contraction; (**b**) thermal expansion of isotropic materials.

**Figure 2 micromachines-11-01069-f002:**
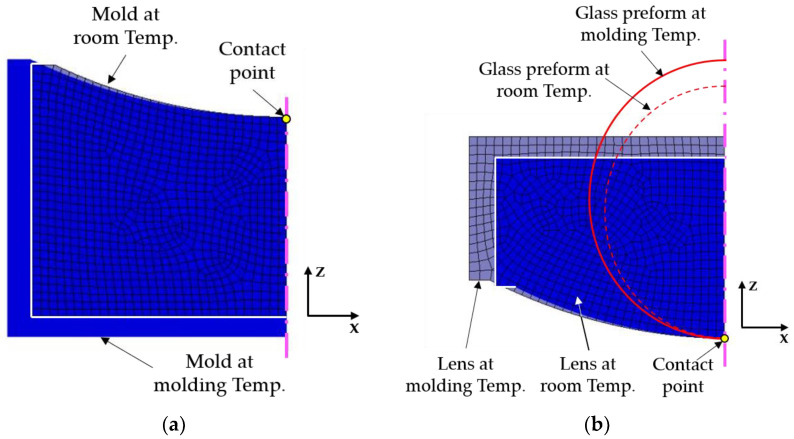
Schematic diagram of expansion and contraction in the precision glass molding (PGM) process. (**a**) Lower mold and (**b**) lens.

**Figure 3 micromachines-11-01069-f003:**
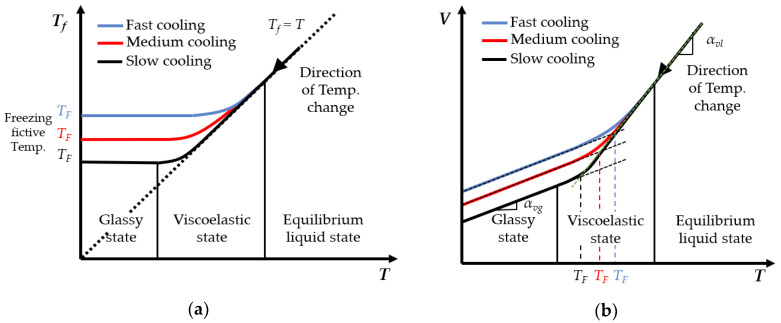
Schematic diagram of fictive temperature and volume change of the glass during the cooling stage in the PGM process. (**a**) Fictive temperature vs. actual temperature. (**b**) Volume change of glass vs. actual temperature.

**Figure 4 micromachines-11-01069-f004:**
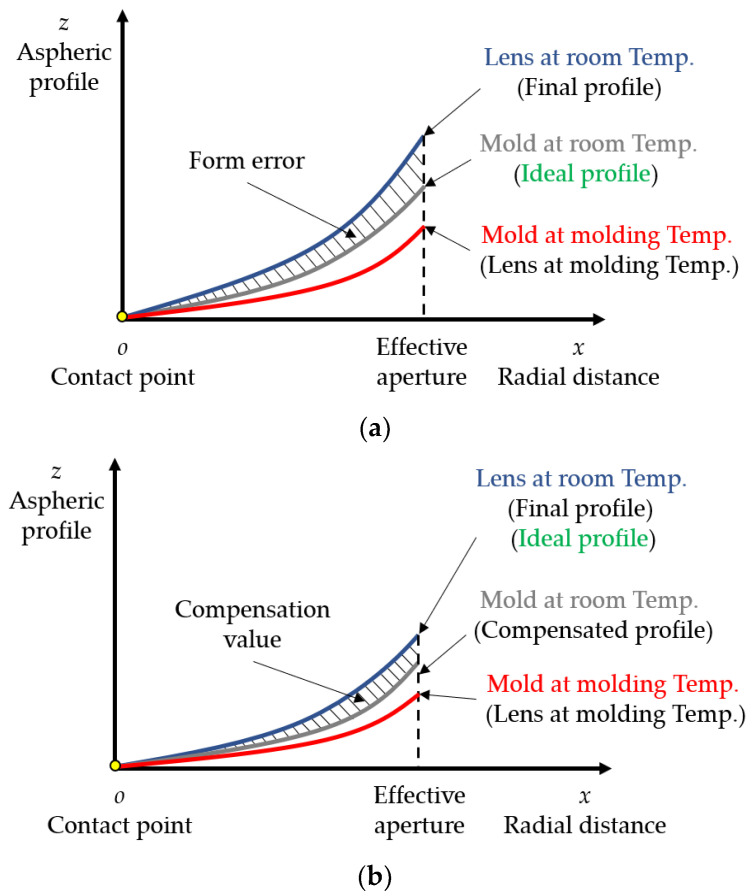
Schematic diagram of the mold and lens surface profiles change in the PGM process. (**a**) Uncompensated mold and (**b**) compensated mold.

**Figure 5 micromachines-11-01069-f005:**
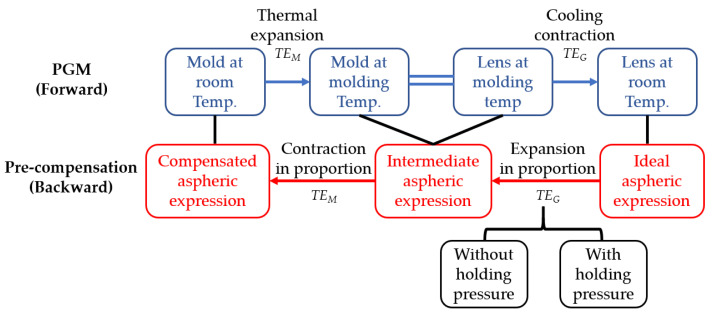
Logic diagram of the mold pre-compensation method based on mathematical analysis.

**Figure 6 micromachines-11-01069-f006:**
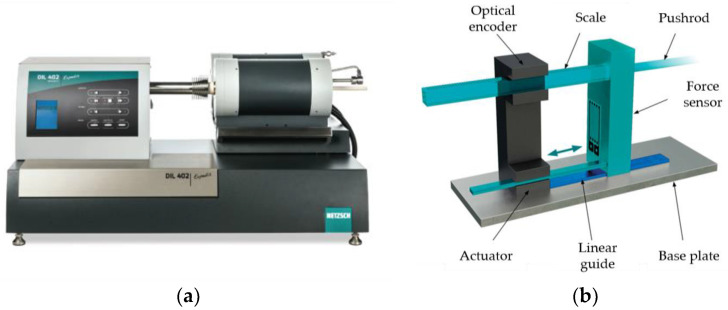
Measuring equipment and principle of thermal expansion curve. (**a**) Measuring equipment and (**b**) schematic diagram of measurement principle.

**Figure 7 micromachines-11-01069-f007:**
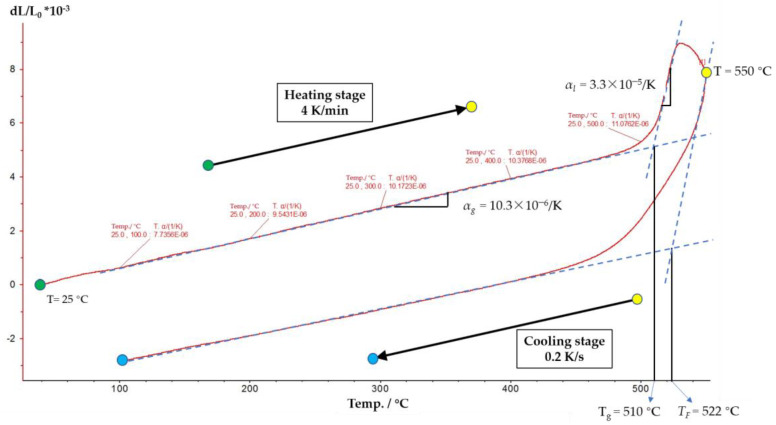
Thermal expansion curve measurement result.

**Figure 8 micromachines-11-01069-f008:**
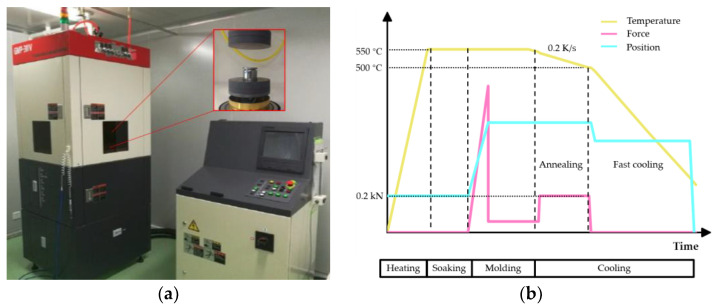
Schematic diagram of PGM process. (**a**) Single station glass molding machine (**b**) PGM process conditions.

**Figure 9 micromachines-11-01069-f009:**
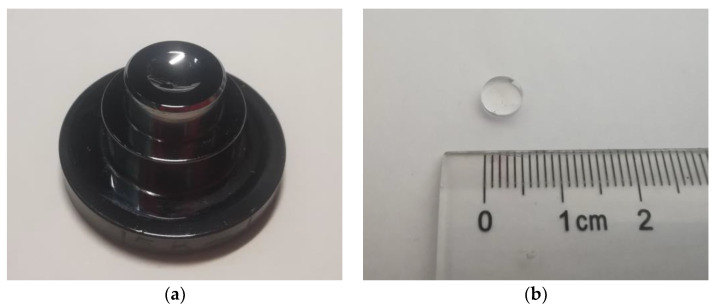
The mold used in PGM experiment and molded glass lens. (**a**) Fine grinding mold and (**b**) molded glass lens.

**Figure 10 micromachines-11-01069-f010:**
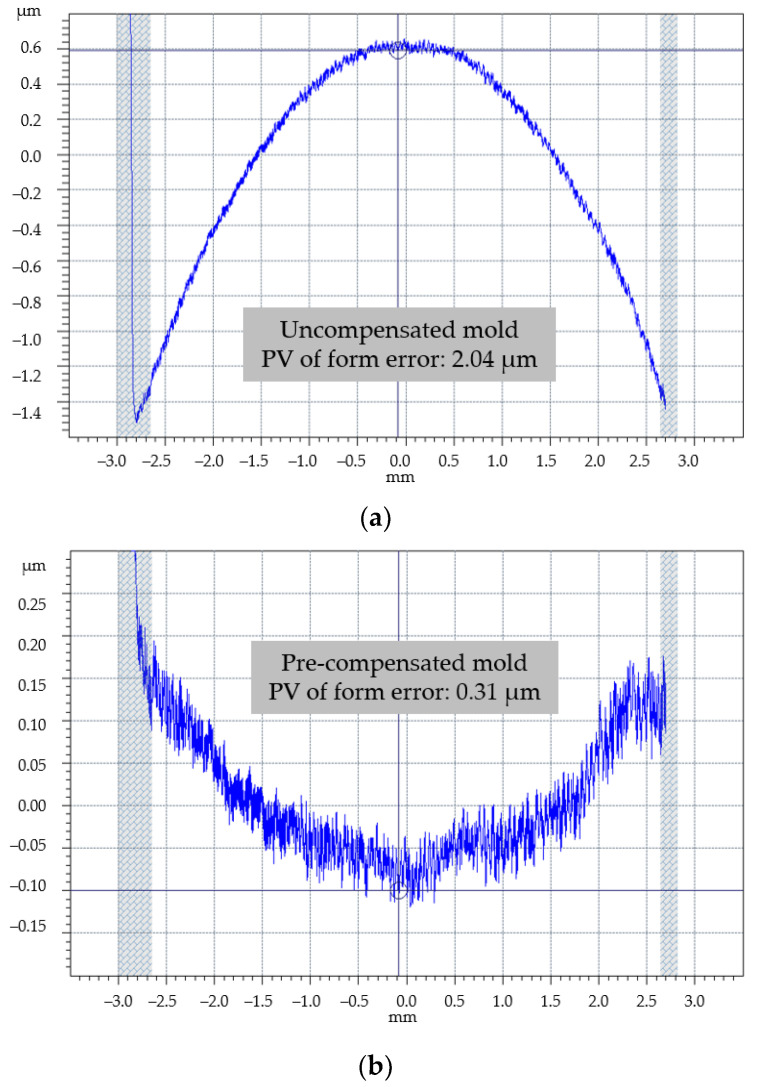
Peak valley (PV) values of aspheric surface form errors of the glass lenses molded by (**a**) uncompensated mold and (**b**) pre-compensated mold.

**Table 1 micromachines-11-01069-t001:** Aspheric coefficients of the ideal lens.

R	K	A_4_	A_6_	A_8_	A_10_
5.7829	−0.9988	2.7653 × 10^−4^	1.0980 × 10^−6^	−1.8716 × 10^−10^	−4.7138 × 10^−14^

**Table 2 micromachines-11-01069-t002:** Aspheric coefficients of the compensated mold.

R	K	A_4_	A_6_	A_8_	A_10_
5.7985	−0.9988	2.7430 × 10^−4^	1.0833 × 10^−6^	−1.8367 × 10^−10^	−4.6008 × 10^−14^

**Table 3 micromachines-11-01069-t003:** Ultra-precision grinding machining parameters for tungsten carbide molds.

Parameter List	Parameter Value	
	Crude Grinding	Fine Grinding
Grinding wheel radius	3.985 mm	3.993 mm
Rounded corner radius	0.198 mm	0.205 mm
Abrasive grain size	#325	#2000
Grinding wheel speed	45,000 rpm	45,000 rpm
Workpiece speed	157 rpm	157 rpm
Cutting depth	2 μm	0.5 μm
Feed rate	2 mm/min	0.5 mm/min

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
