# Peer review of "Pre-Compensation of Mold in Precision Glass Molding Based on Mathematical Analysis"

_micromachines, 2020, doi:10.3390/mi11121069_

Round 1
Reviewer 1 Report
Overall, the paper is well organized and the writing style is acceptable. A minor modification is needed before the publication:
- In Eq (9), what does αM represent?
- In Eq (10), αl and αg represnt?
- On page 7, line 210, changing "at a very slow rate during manufacture" to "at a very slow rate during the fabrication".
Reviewer 2 Report
This manuscript describes pre-compensation technique based on mathematical analysis for glass molding. This reviewer decide to major revision before reconsidering to publish this manuscript. Some reasons can be found in bullet points below:
- The similarity index (plagiarism) is about 7% (after removal the references). Some similarity sentences should be modified. Especially, in abstract, Introduction and theoretical section, there are some overlapping sentences which are similar with the published information elsewhere.
- The introduction is relatively short. The state of art of the glass molding (similar topic) is insufficient.
- In theoretical section, Fig. 3 seems not novel theory, reference could be added.
- In theoretical section, those Equations (Eq. 1,2,3…) are not clear how to obtain. Reference could be added.
- In result and discussion section, it is very unclear how to obtain aspheric coefficient (Table 1 and Table 2). This coefficient is very critical to determine ideal aspheric lens. And it should be repeatable if other readers implement this method.
- The validation is doubtful. Why do the authors only measure peak valley? Why do not include the curvature or radius of the glass lens, for example. Furthermore, the profile in Fig. 9 has different curve profile orientation. Uncompensated mold profile goes up, and pre-compensated mold goes down. And how many experiments have been conducted? Is it repeatable?
- How to validation your mathematical model? How about if some readers conduct with different parameter of the experiments. The authors must add some experimental results with your mathematical model.
